# The Role of Molecular Imaging in Patients with Brain Metastases: A Literature Review

**DOI:** 10.3390/cancers15072184

**Published:** 2023-04-06

**Authors:** Luca Urso, Elena Bonatto, Alberto Nieri, Angelo Castello, Anna Margherita Maffione, Maria Cristina Marzola, Corrado Cittanti, Mirco Bartolomei, Stefano Panareo, Luigi Mansi, Egesta Lopci, Luigia Florimonte, Massimo Castellani

**Affiliations:** 1Department of Nuclear Medicine PET/CT Centre, S. Maria della Misericordia Hospital, 45100 Rovigo, Italy; 2Nuclear Medicine Unit, Fondazione IRCCS Ca’ Granda, Ospedale Maggiore Policlinico, 20122 Milan, Italy; 3Nuclear Medicine Unit, Oncological Medical and Specialist Department, University Hospital of Ferrara, 44124 Cona, Italy; 4Department of Translational Medicine, University of Ferrara, Via Aldo Moro 8, 44124 Ferrara, Italy; 5Nuclear Medicine Unit, Oncology and Haematology Department, University Hospital of Modena, 41125 Modena, Italy; 6Interuniversity Research Center for the Sustainable Development (CIRPS), 00152 Rome, Italy; 7Nuclear Medicine Unit, IRCCS—Humanitas Research Hospital, Via Manzoni 56, 20089 Rozzano, Italy

**Keywords:** brain metastases, positron emission tomography, PET, [^18^F]-FDG, [^18^F]-FET, [^18^F]-DOPA, [^11^C]-MET, radionecrosis

## Abstract

**Simple Summary:**

Brain metastases have an important clinical impact, particularly in terms of treatment and quality of life for cancer patients. Despite the fact that MRI is the imaging modality of choice, we explored the current role of molecular imaging in the context of brain metastases. In addition, we illustrated the potential new application of PET imaging in the future, thanks also to the development of novel targeted therapies, image analysis software, and hybrid acquisition systems (e.g., PET and MRI).

**Abstract:**

Over the last several years, molecular imaging has gained a primary role in the evaluation of patients with brain metastases (BM). Therefore, the “Response Assessment in Neuro-Oncology” (RANO) group recommends amino acid radiotracers for the assessment of BM. Our review summarizes the current use of positron emission tomography (PET) radiotracers in patients with BM, ranging from present to future perspectives with new PET radiotracers, including the role of radiomics and potential theranostics approaches. A comprehensive search of PubMed results was conducted. All studies published in English up to and including December 2022 were reviewed. Current evidence confirms the important role of amino acid PET radiotracers for the delineation of BM extension, for the assessment of response to therapy, and particularly for the differentiation between tumor progression and radionecrosis. The newer radiotracers explore non-invasively different biological tumor processes, although more consistent findings in larger clinical trials are necessary to confirm preliminary results. Our review illustrates the role of molecular imaging in patients with BM. Along with magnetic resonance imaging (MRI), the gold standard for diagnosis of BM, PET is a useful complementary technique for processes that otherwise cannot be obtained from anatomical MRI alone.

## 1. Introduction

Brain metastases (BM) are the most frequent malignant lesions of the central nervous system (CNS) with an incidence four times higher than that of primary brain tumors. BM are observed in 20% to 40% of patients with cancer and the malignant tumors that most frequently metastasize to the brain are lung cancer (>50%), breast cancer (15–25%), malignant melanoma (5–20%) or cancers of unknown primary origin (15%). Magnetic resonance imaging (MRI) is considered the gold standard imaging technique for detecting secondary brain tumors, due to its soft-tissue contrast with high-resolution delineation of tissue anatomy [1]. Indeed, MRI is routinely used for determining the completeness of BM resection when performed within 72 h after surgery, for planning stereotactic radiosurgery, and for response assessment after administration of systemic therapy [2,3,4]. One of the main limitations for conventional MRI is related to the difficulty of discerning between true progression and therapy-related changes, as occurs in pseudo-progression or radionecrosis [5]. In this setting, despite advanced MRI imaging techniques such as perfusion-weighted imaging, spectroscopy, diffusion-weighted imaging, or chemical exchange saturation transfer, all of which have shown promise encouraging results, large multicenter clinical trials are necessary to determine their clinical value [6,7].

Positron Emission Tomography (PET) with Computed Tomography (CT) or MRI (PET/CT or PET/MRI) represents an additional diagnostic tool in neuro-oncological imaging, allowing molecular and metabolic characterization of brain lesions along with anatomical information. [^18^F]F-fluorodeoxyglucose ([^18^F]-FDG) is certainly the most widely used radiotracer in clinical practice, but the high uptake of healthy brain tissue, particularly gray matter, limits its use in the diagnosis of brain tumors due to low tumor-to-background contrast. At the moment, [^18^F]-FDG PET/CT appears useful for staging and for assessment of treatment in patients with primary CNS lymphoma [8,9]. Conversely, radiolabeled amino acid PET tracers are of great interest in brain tumor imaging, as they show a low physiological brain uptake with a better tumor-to-background ratio (TBR) and, potentially, a better diagnostic accuracy. [^11^C]C-methyl-methionine ([^11^C]-MET), O-(2-[^18^F]F-fluoroethyl)-L-tyrosine ([^18^F]-FET), and 3,4-dihydroxy-6-[^18^F]F-L-phenylalanine ([^18^F]-DOPA) are the three most widely used amino acid tracers in clinical practice [10]. They play a key role in primary staging of brain lesions and in biopsy or radiation therapy planning. Moreover, amino acid PET imaging could help to evaluate response to therapy (in particular after radiation therapy), contributing to the differentiation between treatment-related changes and true progression [11,12]. The use of amino acid PET radiotracers in the evaluation of BM has been recently recommended by the international “Response Assessment in Neuro-Oncology” (RANO) working group [13]. Furthermore, the combined use of PET and MRI is widespread in clinical practice and might help oncologists to improve the management of patients with BM, although this combination still has some relevant technical issues [14].

In this narrative review, we provide an overview of the current role of molecular imaging in patients with BM and briefly present future perspectives with new PET radiotracers, the role of radiomics and the newer targeted therapies. Finally, we open a suggestion for a potential application of theranostic approaches in BM, although clinical data are still missing.

## 2. Methods

This review is based on a selective literature search carried out in PubMed up to December 2022. The search string was (positron emission tomography[MeSH Terms]) AND (brain metastas*[Title/Abstract]).

Overall, our search identified 302 articles. Two authors (E.B. and A.N.) independently reviewed abstracts identified with this search, while a third author (A.C.) was consulted in case of discrepancies. Selected articles were examined in full, processed, summarized, and described in the following paragraphs, according to their relevance and adherence to the topic. 

Inclusion criteria were: original articles, English language, clinical trials (randomized, prospective or retrospective); while exclusion criteria were: editorials, letters, case reports, series including fewer than 5 patients, and absence of peer review. 

## 3. [^18^F]-FDG 

As mentioned previously, due to the high glucose metabolism in the brain, [^18^F]-FDG PET presents moderate sensitivity and specificity for characterizing brain tumors, including BM. Furthermore, inflammatory lesions as well as non-tumor tissues may show a false positive increase in FDG uptake. As a matter of fact, several papers have repeatedly demonstrated that the diagnostic accuracy of [^18^F]-FDG PET regarding brain scans for the detection of BM is inferior to MRI, which is more accurate especially for smaller lesions [15,16,17,18,19]. A meta-analysis including 941 lung cancer patients showed a sensitivity of 77% for gadolinium-enhanced MRI compared to 21% for [^18^F]-FDG PET/CT for detecting BM [20]. Recently, Oldan et al. [21] reviewed [^18^F]-FDG PET/CT images from 212 melanoma patients on whom brain MRI was also performed. They demonstrated that [^18^F]-FDG PET/CT was able to detect BM from melanoma generally greater than 3 cm. Moreover, Lee et al. [22] aimed to compare BM metabolic [^18^F]-FDG uptake between non-small cell lung cancer (NSCLC) and small cell lung carcinoma (SCLC) in 48 patients who were previously staged with brain MRI. They found that [^18^F]-FDG accumulation in metastatic brain lesions was variable, with both hypermetabolic (67.3%) and hypometabolic (32.7%) measurements in comparison to gray matter uptake. Among them, BMs from NSCLC were more frequently hypermetabolic than those from SCLC (80% and 26.7%, respectively, *p* < 0.01). 

Few studies have explored the role of [^18^F]-FDG PET/CT in differentiating between BM and primary brain tumors. Meric et al. [23] retrospectively examined whether the use of metabolic parameters could improve the diagnostic ability of [^18^F]-FDG PET/CT to differentiate the nature of brain masses. Among them, tumor SUVmax to ipsilateral cortex SUVmax (Tmax:WMimax) ratio proved to be the most accurate parameter for differential diagnosis between glioma and BM, while SUVmax discriminated well between CNS lymphoma and both glioma and BM. The latter was confirmed also by Purandare et al. [24], who showed that CNS lymphomas have higher metabolic activity, expressed by SUVmax alone as well as ratios of tumor SUVmax to those of the contra-lateral cortex (T/C) and white matter (T/Wm), compared to glioblastoma and BM, whereas there was no difference between glioblastoma and BM.

One of the main challenges for neuro-oncologists is to discriminate between treatment-related changes (i.e., pseudoprogression) and disease progression, which may affect clinical management for selected patients, supporting the need or not for further therapeutic actions. 

In 2006 Wang et al. [25] analyzed whether [^18^F]-FDG PET/CT could differentiate tumor recurrence from radiation reaction in 78 BM and 39 primary brain tumors treated with radiotherapy. The positive predictive value was 96%, while the negative predictive value was 55.6%, suggesting [^18^F]-FDG PET/CT as a valuable tool in the detection of tumor recurrence, especially from lung cancer. Another study by Torrens et al. [26] assessed the role of [^18^F]-FDG PET/CT co-registered with MRI for identifying necrosis from progression after gamma knife radiosurgery. This study proved that, despite 6 out of 27 cases returning false negatives, co-registration of [^18^F]-FDG PET with MRI improved the accuracy of [^18^F]-FDG PET/CT interpretation with a sensitivity of 64% and a specificity of 100%. The main limit of this study was the sole use of a visual evaluation of the lesions, without any quantitation of lesion uptake. On the other hand, Horky et al. [27] tried to evaluate the potential use of dual phase [^18^F]-FDG PET/CT to differentiate recurrence from post-treatment necrosis in patients treated with radiotherapy for BM. They calculated lesion SUVmax (L SUVmax) and ratios of L SUVmax to grey matter SUVmax (L/GM) at early and late time points and the change between the two phases. Lesion SUV values, both at early and late evaluation, did not differentiate between recurrence and necrosis, while percentage change in L/GM ratios provided higher sensitivity, specificity, and accuracy when compared to single time point examination. In addition, a prospective trial by Hatzoglou et al. [28] assessed the effectiveness of [^18^F]-FDG PET/CT and dynamic contrast-enhanced MRI (DCE-MRI) in differentiating tumor progression from radiation injury in 53 patients (29 gliomas and 24 BM) with indeterminate enhancing on conventional MRI. Although lesion TBR was a significant predictor of progression with a sensitivity of 68% and a specificity of 82%, plasma volume (Vp) ratio derived from DCE-MRI had the highest accuracy with 92% sensitivity and 77% specificity. More recently, a retrospective analysis by Leiva-Salinas et al. [29] aimed to determine whether [^18^F]-FDG-PET/MRI could be predictive of local disease control following stereotactic radiosurgery in patients with BM. TBR was significantly associated with local tumor control, with an AUC of 0.67, proposing [^18^F]-FDG as a biomarker of response. Table 1 summarizes the main characteristics of [^18^F]-FDG studies in patients with BM.

## 4. Amino Acid Radiotracers

Recently, the RANO working group has recommended the introduction of amino acid PET radiotracers in the clinical use of patients with brain metastasis [13]. Indeed, due to the well-known MRI limitations, amino acid PET is particularly useful for discriminating between tumor recurrence and treatment-induced changes [30]. [^11^C]-MET, [^18^F]-FET, and [^18^F]-DOPA represent the three most widely used amino acid PET radioligands. Independently from the integrity of the blood–brain barrier, amino acid radiotracers are internalized into tumor cells by L1- and L2- transporters that are overexpressed in gliomas and BM due to increased protein synthesis [31].

### 4.1. [^11^C]-MET

[^11^C]-MET was the first amino acid tracer to be developed, even though its use is limited only to centers with an on-site cyclotron due to its short half-life of only 20 min.

Most studies investigated the role of [^11^C]-MET for treatment monitoring of BM [32,33,34] (Figure 1). Minamimoto et al. [32] compared visual and quantitative analysis to differentiate between radionecrosis and tumor recurrence in 73 brain lesions (31 gliomas, 42 BM). As a result, no significant differences between quantitative assessment and visual analysis were found. However, recently, Govaerts et al. [33] showed that all metabolic quantitative parameters, specifically tumor-to-normal tissue (T/N) ratios such as SUVmean, SUVmax, SUVpeak, T/Nmean, T/Nmax-mean and T/Npeak-mean, TLMM (the product of metabolic tumor volume and SUVmean), were significantly higher in tumor progression than in lesions with treatment-related changes. Of note, SUVmax showed the highest diagnostic performance with an AUC of 0.834 and a cut-off value of 3.29. Sensitivity, specificity, and positive and negative predictive values were 78.57%, 70.59%, 74.32%, and 75.25%, respectively.

In one of the few prospective studies, including 32 patients with 37 BM, Yomo et al. [34] demonstrated that TBRmax of 1.40 showed 0.84 AUC, 82% sensitivity and 75% specificity for discriminating patients with radionecrosis and local recurrence.

The correct definition of tumor volume for radiotherapy is another field of application for amino acid PET radiotracers, in order to deliver the higher dose to the tumor and to preserve surrounding normal brain. First, Matsuo et al. [35] compared the volumes respectively defined by [^11^C]-MET and MRI for radiation therapy planning in 19 patients with 95 BM. For lesions >0.5 mL with MRI, gross tumor volume (GTV) on PET imaging was significantly larger than that on MRI, demonstrating a substantial impact on radiation therapy planning by [^11^C]-MET. Conversely, Momose et al. [36] evaluated the clinical impact of [^11^C]-MET in patients who underwent stereotactic radiosurgery with gamma knife after a previous irradiation. Treatment planning was based on [^11^C]-MET PET/MRI fused images for 34 patients, and on MRI images for 54 patients. The irradiated volume was smaller in the PET/MRI group than in the MRI group. Moreover, the [^11^C]-MET PET/MRI group was associated with a longer overall survival (OS) compared to the MRI group (18.1 vs. 8.9 months, *p* < 0.01), and the [^11^C]-MET PET/MRI group was an independent predictor for OS at multivariate analysis (Odds Ratio 0.54, *p* = 0.02).

Furthermore, [^11^C]-MET diagnostic accuracy was compared with other radiotracers in two studies [2,9]. Rottenburger et al. [37] evaluated the diagnostic performance of [^11^C]-MET and [^11^C]-choline PET/CT for detecting BM. [^11^C]-choline showed higher lesion-to-normal brain (LNR) values compared to [^11^C]-MET (6.6 vs. 1.5, *p* = 0.007), although the study included only eight patients. On the other hand, Tran et al. [38] assessed tumor progression and radionecrosis with two different radiotracers in a feasibility study with five patients who underwent stereotactic radiosurgery. They used [^11^C]-MET and [^11^C]-PBR28, a translocator protein (TSPO) PET target usually associated with microglia activation and neuro-inflammation. While [^11^C]-MET confirmed tumor progression in seven out of seven lesions, [^11^C]-PBR28 was accurate in only three out of seven lesions, thus demonstrating a low specificity and being, consequently, an unreliable marker for radionecrosis.

Cicuendez et al. [39] have investigated the relationship between [^11^C]-MET uptake and histological grade in 35 primary brain tumors and 8 BM. The mean tumor/cortex (T/C) ratio values were greater in high grade gliomas (2.7 ± 1) and BM (2.5 ± 0.7) than in benign lesions or low grade gliomas. Furthermore, patients with a T/C ratio below the threshold value of 1.9 had a longer OS (28 vs. 14 months, *p* = 0.01).

### 4.2. [^18^F]-FET

In view of the limits of [^11^C], related to its short half-life, other amino acid radiotracers labelled with [^18^F] have been developed, such as [^18^F]-FET and [^18^F]-FDOPA, which are easier to manage in clinical practice because of their longer half-life (109 min) and the possibility of shipping the radiotracer to other centers without an on-site cyclotron. [^18^F]-FET is an artificial amino acid characterized by a transient incorporation into the cell. Therefore, a dynamic PET/CT acquisition is allowed by its pharmacokinetics, and it is considered useful to characterize brain lesions [40,41].

Unterrainer and colleagues [42] analyzed the characteristic of newly diagnosed and untreated BM by [^18^F]-FET PET. The study included 30 patients with 45 metastases. Forty metastases were [^18^F]-FET-positive with a TBR max > 1.6, while five metastases were classified as FET-negative. Furthermore, there was no significant difference of metabolic parameters (i.e., TBRmax, TBRmean, BTV) among lung, breast, and melanoma [^18^F]-FET-positive lesions. All metastases from lung cancer showed a high [^18^F]-FET uptake, which was independent from lesion size, while a wide uptake variability was observed for melanoma metastases due to the different characteristics of primary tumor. 

Regarding the potential role of [^18^F]-FET PET/CT for differentiating radionecrosis from tumor recurrence, in this latter group, both TBRmax and TBRmean were significantly higher [43,44,45,46]. In particular, cut-off values of 2.15–2.55 for TBRmax and 1.95–1.99 for TBRmean showed an overall good accuracy with a sensitivity and specificity ranging from between 79–86% and 76–86% for the first parameter and 74–86% and 79–90% for the second. Moreover, the time–activity curve (TAC) showed a constantly increasing tracer uptake for radionecrosis (pattern I), whereas recurrent brain metastases had an early peak (≤20 min) followed by either a plateau (pattern II) or a constant descent (pattern III). Finally, by adding kinetic and static [^18^F]-FET PET parameters, an increase in both sensitivity and specificity was observed, with values ranging from 88–95% to 83–91%, respectively (Figure 2).

In comparison with other tracers, Grosu et al. [47] evaluated [^18^F]-FET PET and [^11^C]-MET PET in patients with gliomas and BM. They studied 42 patients (29 gliomas, 13 BM) with suspicious tumor recurrence on MRI images. They found a high correlation between MET and FET uptake in the tumor tissue (rho = 0.84). Both [^18^F]-FET and [^11^C]-MET PET showed a comparable high sensitivity and specificity for tumor tissue, of 91% and 100%, respectively. With the aim of radiation therapy planning, the differences in GTV between [^18^F]-FET and [^11^C]-MET PET were not statistically significant. High sensitivity (91%) and specificity (100%) for both [^18^F]-FET and [^11^C]-MET PET in differentiating residual/recurrent tumors from pseudoprogression were also shown in this study. 

Finally, Gempt et al. [48] compared the congruence of tumor volumes between MRI and [^18^F]-FET PET/CT in 41 patients with BM before neurosurgery. According to their results, tumor volumes only partially overlapped, suggesting that MRI and [^18^F]-FET PET/CT could play a synergic role in tumor extension delineation in metastatic brain lesions.

### 4.3. [^18^F]-DOPA

Similar to methionine, DOPA, an amino acid-related compound, participates directly in protein synthesis after being converted into dopamine by a decarboxylase enzyme. Its transportation is mediated with high affinity by LAT-1 and LAT-2, responsible also for cellular uptake of large neutral or aromatic amino acids [49]. However, LAT-1 needs a cofactor, CD98, to perform its function as an amino acid transporter. As a matter of fact, LAT-1 and CD98 are overexpressed in several tumors, and their expression often correlates with more aggressive histology and worse prognosis. Recently, a French group has demonstrated a significant higher expression of LAT-1 and CD98 in BM than in non-tumoral brain tissue (98.5% vs. 59.7%, *p* < 0.001). Furthermore, LAT-1 overexpression has been correlated with [^18^F]-DOPA uptake (*p* = 0.037), while radionecrosis was associated with low LAT-1 expression and no or low [^18^F]-DOPA uptake (ratio SUVmax lesion/SUVmax striatum 0.75, *p* = 0.003) [31] (Figure 3).

In the differentiation of recurrent BM and radionecrosis, other studies have demonstrated encouraging results, with a sensitivity ranging from 81% to 90% and a specificity ranging from 84% to 96% [30,49,50,51]. In particular, Lizarraga et al. [50] evaluated the role of [^18^F]-DOPA in 32 patients with 83 irradiated BMs. PET/CT scans were evaluated both semi-quantitatively (i.e., lesion-to-striatum and lesion-to-normal brain tissue ratios based on SUVmax and SUVmean values) and visually (4-point score). Visual analysis demonstrated the best accuracy, where a score ≥ 2 determined a sensitivity of 81.3% and a specificity of 84.3%. In addition, [^18^F]-DOPA was the strongest predictor of tumor progression (HR 6.26, *p* < 0.001) among different variables, and negative PET/CT scan was associated with longer time to progression than positive scans (76 vs. 16 months, *p* < 0.001). Moreover, Cicone and colleagues [51] compared [^18^F]-DOPA and perfusion MRI in order to differentiate between radionecrosis and tumor progression in 50 BM after stereotactic radiosurgery. In this setting, TBR (SUVLmax/Bkgrmax) showed the best diagnostic performance with a cut-off value of 1.59 (sensitivity 90%, specificity 92%, and accuracy 91%). On the other hand, rCBV had a lower performance than all metabolic parameters, with a sensitivity, specificity, and accuracy of 87%, 68%, and 76%, respectively. The same Italian group [52] assessed the evolution of radionecrosis in 34 BMs using [^18^F]-DOPA PET/CT every 6 months or yearly along with standard MRI. Semi-quantitative parameters, such as TBR and relative SUV (rSUV), increased significantly over time with local progression, whereas they remained stable in radionecrosis. rSUV showed the best performance, with an accuracy of 94.15% for the optimal cut-off value of 1.92, while variation in the longest tumor dimension measured on contrast-enhanced MRI did not differentiate between radionecrosis and progression. 

Finally, in a recent monocentric clinical trial, Humbert et al. [53] assessed the impact of [^18^F]-DOPA on a multidisciplinary neuro-oncology tumor board in patients with both glioblastoma (*n* = 65) and BM (*n* = 41). Overall, [^18^F]-DOPA changed diagnosis and treatment planning in 39% and 17% of cases, respectively. In patients with BM, the adjunction of [^18^F]-DOPA PET increased the Younden’s index from 0.44 to 0.53, improving the diagnostic accuracy. Table 2 summarizes the main characteristics of studies using amino acid PET radiopharmaceuticals.

## 5. Other Radiotracers

The intrinsic limitations of [^18^F]-FDG-PET in the evaluation of brain lesions have paved the way for the research of new radiotracers with a more favorable TBR in the encephalic district. Among those, Alpha [^11^C]methyl-L-tryptophan ([^11^C]-AMT) dynamic PET has been assessed by Kamson et al. [54] to discriminate glioblastomas from BMs in 36 patients with newly diagnosed brain tumors. The results were encouraging, as the evaluation of AMT uptake could improve pre-treatment differentiation of the two brain malignancies. In particular, among solitary ring-enhancing lesions, which are often a clinical dilemma to solve, [^11^C]-AMT PET showed higher tumor/cortex SUV ratios in glioblastomas than in metastatic tumors, allowing a correct differentiation in more than 90% of cases. Nevertheless, as mentioned above, the use of [^11^C] as radionuclide could limit the widespread use of this compound.

In another prospective trial, Xu et al. [55] evaluated [^18^F]-(2S,4R)-4-fluoroglutamine ([^18^F]-FGln), a metabolic PET indicator for glutamine metabolism, in 14 patients with suspected BM and compared the results with those obtained by [^18^F]-FDG PET and/or ceMRI. [^18^F]-FGln outperformed [^18^F]-FDG PET with a per-lesion detection rate of 81.6% and 36.8%, respectively. At semiquantitative analysis, TBR was significantly better for [^18^F]-FGln (*p* = 0.05). Moreover, [^18^F]-FGln uptake was independent of BM size and presence of peripheral edema and was detected also in FDG-avid extra-cranic metastases (except in liver and bone metastases that may appear as “cold lesions” due to the high surrounding physiological uptake).

A feasibility study with [^18^F]-AlF-NOTA-E[PEG_4_-c(RGDfk)]_2_ ([^18^F]-Alfatide II), an integrin αvβ3 specific PET tracer, was used to image nine patients with BM, after it was successfully tested in five healthy volunteers [56]. [^18^F]-Alfatide II PET showed significant uptake in correspondence of 20/20 lesions in the nine pathological patients, probably due to the increased angiogenesis and integrin expression of BM. Conversely, [^18^F]-FDG-PET and low-dose CT could identify only 10 and 13 lesions, respectively.

Therapy response assessment is another open issue in patients with BM. In the literature, [^18^F]-choline PET/CT has been used for the evaluation of both primary brain tumors and BM [57]. Grkovski et al. [58] tested [^18^F]-choline PET/CT to discriminate recurrence vs. radionecrosis (RN) in 12 patients previously treated with stereotactic radio-surgery for BM. Higher SUVmax values on dynamic preoperative [^18^F]-choline PET/CT could discriminate recurrence vs. RN (*p* = 0.01) and SUVmax > 6 was a negative prognostic factor. However, these results should be handled gingerly as only 2 out of 12 patients had RN. Moreover, two papers investigated [^18^F]-3′deoxy-3′- fuorothymidine ([^18^F]-FLT) PET to assess response to therapy in breast cancer (BC) patients with BM. [^18^F]-FLT is a thymidin analogue reflecting cellular proliferation, thus a reduction in uptake is expected after cyto-reductive treatments. Morikawa et al. [59] used [^18^F]-FLT PET to assess early response to sorafenib and whole-brain radiation therapy (WBRT) treatment with promising results. O’Sullivan et al. [60] compared [^18^F]-FLT PET and contrast-enhanced MRI in the evaluation of response to ANG1005, an experimental conjugate of paclitaxel and Angiopep-2 designed to cross the blood–brain barrier. As a result, the authors report a moderately strong association (Spearman rho > 0.7) between the response obtained with ceMRI and [^18^F]-FLT PET, thus proposing [^18^F]-FLT PET as a potential complementary imaging in patients with BM. Another radiotracer used to assess response to WBRT is [^18^F]-ML-10, a low-molecular-weight PET probe whose uptake reflects in vivo apoptosis. In the paper published by Allen et al. [61] baseline [^18^F]-ML-10 detected every lesion visualized by MRI in 10 patients with BM. A post-radiation therapy (RT) PET scan showed a substantial increase in lesion [^18^F]-ML-10 uptake. Interestingly, the changes in tumor uptake at early [^18^F]-ML-10 PET showed a highly significant correlation with those measured on MRI 6–8 weeks later. Therefore, [^18^F]-ML-10 PET may be a candidate for early imaging of RT-induced cell-death in patients with BM, but large prospective trials are needed to confirm this preliminary hypothesis.

More recently, a Norwegian group compared the artificial amino acid [^18^F]-fluciclovine PET/MRI, also known as anti-1-amino-3-^18^F-fluorocyclobutane-1- carboxylic acid (FACBC), with MRI alone in patients with BM [62]. Despite BM being positive with [^18^F]-fluciclovine, PET/MRI did not significantly improve BM detection compared with MRI alone. However, PET/MRI was able to identify tumor tissue beyond contrast enhancement on MRI.

Recently, 16-alpha-^18^F-fluoro-17-beta-estradiol (FES) PET has been approved for clinical use in France and USA in recurrent or metastatic estrogen receptor (ER)-positive breast cancer. Indeed, [^18^F]-FES PET/CT provides a noninvasive estimation of ER distribution, allowing therefore its application in multiple clinical scenarios, for example: (a) to stage initial metastatic disease; (b) to evaluate ER status when biopsy is difficult or non-diagnostic; (c) to assess endocrine therapy response; (d) to stage invasive lobular carcinoma and low-grade invasive ductal carcinoma which are more likely to present with BM [63,64,65,66].

Table 3 illustrates other PET radiotracers developed for studying BM, some of which are new.

## 6. Future Perspectives

### 6.1. Radiomics

Radiomics represents a new emerging field in the quantitative analysis of medical imaging. It is based on the extraction, by a machine- or deep-learning approach, of quantitative features from images that normally are challenging to detect for the human eye, in order to characterize a pattern of pathology and to find an association with clinical outcomes, or to evaluate response to treatment [67,68,69,70].

A retrospective study by Cao et al. [71] has explored the ability of radiomics features, both for MRI and for [^18^F]-FDG PET/CT, to differentiate between primary brain tumors and BMs. They analyzed 50 patients with glioblastoma and 50 with solitary BM, building three models from MRI, [^18^F]-FDG PET, and their combination. The latter showed the highest AUC compared with MRI or [^18^F]-FDG-PET taken singularly. Therefore, this model could help to differentiate primary brain lesions from BM before surgery.

In order to differentiate between recurrence and radiation injury, Lohmann et al. [72] performed a radiomics analysis of [^18^F]-FET PET/MRI in 52 patients with BM previously treated with radiotherapy. Diagnostic accuracy, sensitivity, and specificity were slightly higher for [^18^F]-FET PET than MRI textural features (83%, 88%, and 75% vs. 81%, 67%, and 90%, respectively). However, the best performance was reached when combining four MRI and one PET radiomics features, which allowed a diagnostic accuracy, sensitivity, and specificity of 89%, 85%, and 96%, respectively. The same group previously combined semi-quantitative metabolic parameters and radiomics from [^18^F]-FET PET in 47 patients with single or multiple BMs treated with radiotherapy [73]. The association of metabolic parameters, such as TBRmax and TBRmean, with textural features increased the diagnostic accuracy for discriminating between progression and treatment-related changes, whereas no improvement was detected when combining radiomics and kinetic parameters.

Finally, Stefano et al. [74] used a radiomics approach to delineate biological target volume (BTV) by [^11^C]-choline PET in 56 BM. Of note, three features, i.e., asphericity, low-intensity run emphasis and complexity, and their combination, showed the best performance to discriminate between responders and non-responders to RT. Moreover, for follow-up evaluation, eight radiomics features demonstrated a sensitivity, specificity, and accuracy of 86.3%, 87.8%, and 86.6%, respectively.

### 6.2. Therapy with Immune Checkpoint Inhibitors

In recent times, treatments with immune checkpoints inhibitors (ICI) and other immunotherapy options are under investigation for patients with primary brain tumors and BM. In parallel, these promising therapeutic agents impose new demands for brain imaging.

Akhoundova et al. [75] evaluated the ability of [^18^F]-FET PET to distinguish pseudo- from real progression in BMs from NSCLC patients treated with radiotherapy and immunotherapy. This study included 53 patients; after radiotherapy 30 patients showed progression of at least one treated metastasis on MRI images. Eleven patients were subjected to [^18^F]-FET PET, which correctly identified 90% of patients with pseudoprogression.

Kebir et al. [45] conducted a small retrospective study to assess whether [^18^F]-FET PET might be valuable for distinguishing pseudoprogression in patients with melanoma BM. Five patients were investigated with [^18^F]-FET PET while under ICI treatment. Four patients with high TBRmax presented true tumor progression, while the other patient with lower TBR demonstrated pseudoprogression. Additionally, TTP (time-to-peak values) were inversely correlated with pseudoprogression.

Immuno-PET, combining antibodies or their fragments with a radiotracer, can target the main protagonists of immune response, such as CD4+ or CD8+ T cells and immune checkpoints, in order to predict therapeutic response and to improve patients’ selection [76]. In a recent phase I/II clinical trial, de Ruijter et al. [77] have shown that radiolabeled [^89^Zr]ZED88082A has the potential to characterize the dynamic changes of CD8+ T cells during ICI, opening the possibility of immuno-PET imaging as a predictive biomarker.

### 6.3. Theranostics

In recent decades, the combination of diagnostics and therapeutics, so-called “theranostics”, has been one of the main advantages in the field of nuclear medicine and precision oncology. Prostate-specific membrane antigen (PSMA) is a trasnmembrane protein over-expressed in prostate cancer but also in several other malignancies, including primary brain tumors and BM [78,79]. As a matter of fact, PET-based PSMA is an example of a theranostics approach which exchanges [^68^Ga], used for diagnostic imaging, with therapeutic beta-emitters such as [^177^Lu] or [^90^Y]. However, while encouraging results have been achieved in prostate cancer patients, the potential use of theranostics in neuro-oncology is still under investigation and needs further robust evidence.

### 6.4. Novel Targeted Therapies and PET Imaging

Mutations on transmembrane protein receptors belonging to the epidermal growth factor receptor (EGFR) family are well known to be associated with the development of numerous malignancies. At the same time, such mutations are suitable targets for therapy with monoclonal antibodies as well as for imaging. [^11^C]erlotinib, [^11^C]PD153035, and [^89^Zr]Zr-DFO-nimotuzumab are PET ligands used for detecting EGFR overexpression, while [^64^Cu]DOTA-trastuzumab and [^89^Zr]pertuzumab are used for studying HER2 overexpression [80,81,82,83,84,85].

## 7. Conclusions

To summarize, our review illustrates the role of molecular imaging in patients with BM. Currently, MRI represents the gold standard for diagnosis of BM, but PET is a valid complementary technique providing important biological findings that cannot be obtained from anatomical MRI alone. Amino acid PET radiotracers are the major representative of nuclear medicine in neuro-oncology, in particular for the diagnosis and assessment of post-treatment changes where results are encouraging. Furthermore, a wide variety of other PET agents have been developed and proposed for investigating different biological processes or for assessing response to newer therapies in patients with BM, including neuropathological validation of imaging findings. Finally, the introduction of hybrid systems, e.g., PET/MRI, may improve the diagnostic process as they allow the acquisition of numerous multimodal imaging parameters in a shorter time, although at higher costs. However, large prospective clinical trials are necessary to confirm such preliminary positive impressions in these cohorts of patients.

## Figures and Tables

**Figure 1 cancers-15-02184-f001:**
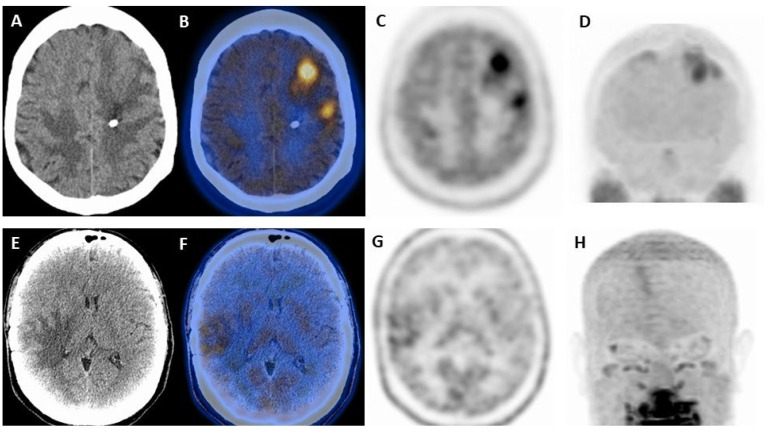
Comparison of two patients with brain metastases investigated with [^11^C]-MET PET/CT. (**A**–**D**) Frontal brain metastases from breast cancer (SUVmax 6.4; normal brain 2.5) confirmed on PET/CT; (**E**–**H**) Radionecrosis in a patient with brain metastasis from melanoma previously treated with radiotherapy (SUVmax 1.97; normal brain 1.27). (**A**,**E**) low-dose CT; (**B**,**F**) fused axial PET/CT; (**C**,**G**) axial [^11^C]-MET PET; (**D**,**H**) MIP (maximal intensity projection) images.

**Figure 2 cancers-15-02184-f002:**
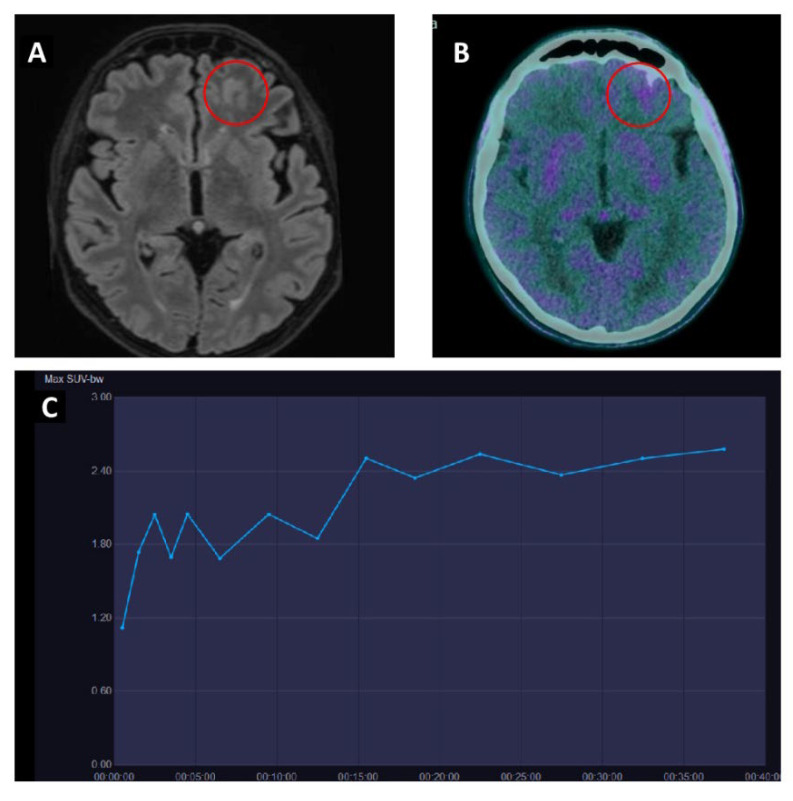
Patient with breast cancer and suspected recurrence of BM after radiotherapy. (**A**) Brain MRI shows an area of hyperintense TR signal on left frontal-orbital cortex (red circle). (**B**) Static [^18^F]-FET PET/CT demonstrated only a slight [^18^F]-FET uptale (SUVmax 2.6; TBRmax 1.8) (red circle). (**C**) Dynamic time-activity curve of [^18^F]-FET PET/CT shows an ascending pattern with a late time to peak (>20 min). PET/CT suggested a treatment related change and patient is still free from disease approximately one year after treatment.

**Figure 3 cancers-15-02184-f003:**
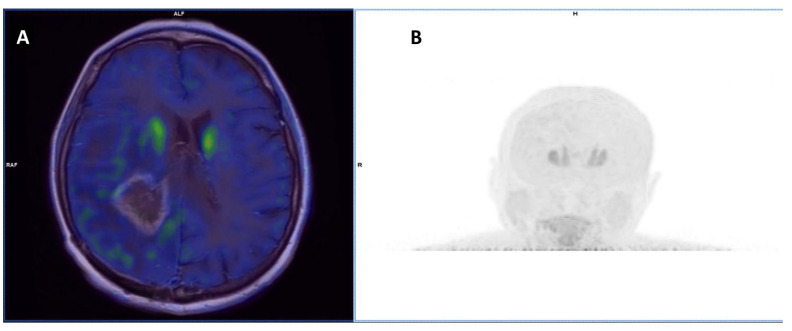
(**A**). [^18^F]-DOPA PET transaxial images fused with diagnostic brain T1 FSE ceMRI in a patient with multiple BM from breast cancer. The patient was previously treated with WBRT followed by surgical excision of the largest BM (right parietal lobe) and stereotactic RT on the surgical bed. Post-RT ceMRI showed a significant size increase in the right periventricular area with ring-enhancement in differential diagnosis between radionecrosis and disease progression. PET showed the absence of correspondent [^18^F]-DOPA uptake, compatible with pseudoprogression. (**B**). [^18^F]-DOPA maximum intensity projection image, showing encephalic physiological distribution of [^18^F]-DOPA; note the symmetric intense striatal uptake.

**Table 1 cancers-15-02184-t001:** Summary of general characteristics of studies with [^18^F]-FDG.

Authors	Year	Study Design	Primary Malignancy	Patients M/F	Aim	Comments
Krüger et al. [15]	2011	P	Lung cancer	104 (77/27)	To compare MRI and PET/CT for diagnosis BM	PET/CT showed a sensitivity of 27%, with a high number of false positive for BM
Bochev et al. [16]	2012	P	Solid neoplasms	2502 (NR)	To assess the role of PET/CT for detecting BM	PET/CT detected BM in 1% of all patients
Manohar et al. [17]	2013	R	Solid neoplasms	5110 (3322/1788)	To assess the role of PET/CT for detecting BM	PET/CT detected BM in only 0.7% of cases
Nia et al. [18]	2017	R	NSCLC	227 (NR)	To assess the role of follow-up PET/CT for detecting BM	Only 5/227 patients were found to have BM
Saito et al. [19]	2021	R	T1-T2 N0 NSCLC	466 (272/194)	To assess the frequency of BM	Screening of brain by PET/CT is unnecessary in patients with early stage NSCLC
Li et al. [20]	2017	Meta-analysis	Lung cancer	941 (NR)	To compare MRI and PET/CT for diagnosis BM	Gadolinium-enhanced MRI had higher sensitivity than PET/CT
Oldan et al. [21]	2020	R	Melanoma	212 (NR)	To evaluate at what size BM are detectable by PET/CT	Lesions over about 2 cm were detectable by PET/CT
Lee et al. [22]	2008	R	Lung cancer	48 (31/17)	To compare FDG uptake between NSCLC and SCLC BM	NSCLC BM were more frequently hypermetabolic than those from SCLC
Meric et al. [23]	2015	R	BM, CNS lymphomas, gliomas	76 (37/39)	To characterize the nature of brain masses	SUVmax and Tmax:Wmimax seem useful parameters to discriminate brain masses
Purandare et al. [24]	2017	R	GBM, CNS lymphoma, BM	106 (70/36)	To characterize the nature of brain masses	CNS lymphomas showed higher metabolic activity than GBM and BM
Wang et al. [25]	2006	R	BM, gliomas	117 (58/59)	To differentiate between recurrence and radionecrosis	PET/CT demonstrated: PPV 96% NPV 56%
Torrens et al. [26]	2016	R	BM, glioblastoma	16 (11/5)	To differentiate between recurrence and radionecrosis co-registering PET/CT and MRI	PET/MRI co-registration determined: Sensitivity 65% Specificity 100%
Horky et al. [27]	2011	R	Solid neoplasms	32 (10/22)	Dual phase PET/CT to differentiate recurrence from radionecrosis	Variation of L/GM > 0.19 between early and delayed: Sensitivity 955% Specificity 100% Accuracy 96%
Hatzoglou et al. [28]	2016	P	BM, gliomas	53 (35/18)	To differentiate between recurrence and radionecrosis using PET/CT and DCE MRI	Vp ratio = 2.1 showed highest accuracy: Sensitivity 92% Specificity 77%
Leiva-Salinas et al. [29]	2019	R	BM	85 (37/48)	To determinate if PET/MRI predicts recurrence after radiosurgery	Relative SUV = 1.75: Sensitivity 87% Specificity 32%

Abbreviations: BM: brain metastases; CNS: central nervous system; DCE: dynamic contrast enhancement; L/GM: lesion/gray matter ratio; NPN: negative predictive value; NR: not reported; NSCLC: non small cell lung cancer; R: retrospective; P: prospective; PPV: positive predictive value; Vp: plasma volume.

**Table 2 cancers-15-02184-t002:** Summary of general characteristics for studies with amino acids radiotracers.

Authors	Year	Study Design	RP	Primary Malignancy	Patients M/F	Aim	Comments
Minamoto et al. [32]	2015	R	[^11^C]-MET	BM and gliomas	70 (38/32)	To differentiate between recurrence and radionecrosis	Visual analysis was comparable to quantitative assessment by L/Nmax and L/Nmean
Govaerts et al. [33]	2021	R	[^11^C]-MET	Solid neoplasms	26 (13/13)	To differentiate between recurrence and radionecrosis	SUVmax of 3.9 was the best parameter: AUC = 0.834 sensitivity 78.6% specificity 70.6% PPV 74.3% NPV 75.3%
Yomo et al. [34]	2017	P	[^11^C]-MET	Solid neoplasms	32 (19/13)	To differentiate between recurrence and radionecrosis	LNR of 1.40 showed: AUC 0.84 sensitivity 82% specificity 75%
Matsuo et al. [35]	2009	P	[^11^C]-MET	Solid neoplasms	19 (14/5)	To delineate and to compare target volumes with MRI	Tumor volume on PET imaging was significantly larger than that on MRI for lesions >0.5 mL
Momose et al. [36]	2014	R	[^11^C]-MET	NR	88 (48/40)	To differentiate between recurrence and radionecrosis	[^11^C]-MET-PET was predictive for longer OS after stereotactic radiosurgery
Rottenburger et al. [37]	2011	P	[^11^C]-MET, [^11^C]-choline	Solid neoplasms	8 (NR)	To compare [^11^C]-MET and [^11^C]-choline PET	[^11^C]-choline showed a higher LNR
Tran et al. [38]	2020	P	[^11^C]-MET, [^11^C]PBR28	Melanoma, NSCLC	5 (3/2)	To compare [^11^C]-MET and [^11^C]PBR28 PET	[^11^C]PBR28 was not a valid biomarker to detect radionecrosis
Cicuendez et al. [39]	2015	P	[^11^C]-MET	NR, gliomas	43 (24/19)	To evaluate [^11^C]-MET uptake and relationship with histopathological grade	T/C was higher in BM and high grade gliomas; T/C < 1.9 was associated with longer OS
Unterrainer et al. [42]	2017	R	[^18^F]-FET	Solid neoplasms	30 (NR)	To evaluate the uptake characteristics of untreated BM	All BM > 1cm were [^18^F]-FET positive
Galldiks et al. [43]	2012	P	[^18^F]-FET	Solid neoplasms	31 (5/26)	To differentiate between recurrence and radionecrosis	TBRmax of 2.55: AUC 0.822 sensitivity 79% specificity 76% TBRmean of 1.95: AUC 0.851 sensitivity 74% specificity 90%
Ceccon et al. [44]	2017	R	[^18^F]-FET	Solid neoplasms	62 (14/48)	To role of dynamic PET scan to differentiate recurrence from radiation injury	TBRmean > 1.95 + a slope < 0.37 SUV/ h: accuracy 88% sensitivity 83% specificity 93%
Kebir et al. [45]	2016	R	[^18^F]-FET	melanoma	5 (NR)	To evaluate pseudoprogression in patients treated with ICI	TBRmax was higher in patients with true progression (5.4 vs. 2.5), as well as time to peak was significantly shorter (17 min vs. 45 min)
Romagna et al. [46]	2016	R	[^18^F]-FET	Solid neoplasms	22 (11/11)	To differentiate between recurrence and radionecrosis	TBRmax of 2.15 and TBRmean of 1.95: AUC 0.84 sensitivity 86% specificity 79% TBRs + decreasing TAC: AUC 0.79 sensitivity 91% specificity 83%
Grosu et al. [47]	2011	P	[^18^F]-FET, [^11^C]-MET	Solid neoplasms, gliomas	42 (NR)	To compare [^18^F]-FET and [^11^C]-MET uptake in gliomas and BM; To compare volumes between PET and MRI	[^18^F]-FET and [^11^C]-MET strongly correlated; Both radiotracers: sensitivity 91% specificity 100%
Gempt et al. [48]	2015	R	[^18^F]-FET	Solid neoplasms	41 (NR)	To delineate and to compare target volumes with MRI	Tumor volumes by [^18^F]-FET and MRI were only partially overlapped
Papin-Michault [31]	2016	R	[^18^F]-DOPA	Solid neoplasms, non-tumoral tissue	67 BM 53 control	LAT-1 and CD68 expression in BM	LAT-1 expression level and [^18^F]-DOPA uptake were significantly correlated
Lizarraga et al. [50]	2014	R	[^18^F]-DOPA	Solid neoplasms	32 (26/6)	To differentiate between recurrence and radionecrosis	Visual scoring ≥ 2: sensitivity 81% specificity 84%
Cicone et al. [51]	2015	R	[^18^F]-DOPA	Solid neoplasms	42 (NR)	To differentiate between recurrence and radionecrosis and to compare with MRI	SUVLmax/Bkgrmax of 1.59: sensitivity 90% specificity 92%
Cicone et al. [52]	2021	P	[^18^F]-DOPA	Solid neoplasms	30 (13/17)	To characterize the long-term metabolic evolution of radionecrosis	rSUV of 1.92: sensitivity 90% specificity 96%
Humbert et al. [53]	2019	P	[^18^F]-DOPA	Solid neoplasms, glioblastoma	106	To evaluate the impact of [^18^F]-DOPA on the therapeutic decision	For suspicions of tumor recurrence, [^18^F]-DOPA improved diagnostic accuracy for both BM and glioblastomas

Abbreviations: BM: brain metastases; CNS: central nervous system; L/N: lesion/normal brai ratio; NR: not reported; R:retrospective; P: prospective; rSUV: relative standardize uptake value; T/C: tumor-to-cortex ratio; TBR: tumor-to-background ratio.

**Table 3 cancers-15-02184-t003:** Summary of general characteristics of studies with other radiotracers.

Authors	Year	Study Design	RF	Primary Malignancy	Patients (M/F)	Aim	Comments
Kamson et al. [54]	2013	P	[^11^C]-AMT	Solid neoplasms, glioblastoma	36 (20/16)	To discriminate between BM and glioblastomas	BM had lower tumoral SUVs, lower mean tumor/cortex SUVratio, and tumor/cortex VD′-ratio
Xu et al. [55]	2018	P	[^18^F]-FGln	Solid neoplasms, gliomas	14 (7/7)	To compare [^18^F]-Fgln and [^18^F]-FDG	Detection rates for BM [^18^F]-Fgln 82% [^18^F]-FDG 37%
Yu et al. [56]	2015	P	[^18^F]-Alfatide II	Solid neoplasms, gliomas	9 (5/4)	To compare [^18^F]-Alfatide II and [^18^F]-FDG	All 20 brain lesions were visualized by [^18^F]-Alfatide II, while only 10 by [^18^F]-FDG, and 13 by CT.
Grkovski et al. [58]	2020	P	[^18^F]-choline	Solid neoplasms	14 (NR)	To evaluate [^18^F]-choline uptake correlation from surgical samples with pathologic evidence of recurrent tumor	Surgical samples with viable tumor had higher uptake than those without tumor, although inflammation and gliosis also increase the uptake
Morikawa et al. [59]	2021	P	[^18^F]-FLT	Breast	15 (NR)	To assess early response to sorafenib and whole-brain radiation therapy	[^18^F]-FLT seems a valid imaging tool for early response assessment
O’Sullivan et al. [59]	2016	P	[^18^F]-FLT	Breast	10 (NR)	To evaluate therapy response	A total of 52% of target lesions showed a reduction in [^18^F]-FLT SUV ≥20% after treatment
Allen et al. [61]	2012	P	[^18^F]-ML-10	Solid neoplasms	10 (NR)	To evaluate therapy response after radiation therapy	High correlation between early changes on [^18^F]-ML-10 PET and later changes on MRI
Øen et al. [62]	2022	R	[^18^F]-fluciclovine	Solid neoplasms	18 (11/7)	To compare diagnostic accuracy for tumor recurrence between PET/MRI and MRI alone	PET volumes correlated and were comparable in size with those from MRI, but were only partially congruent

Abbreviations: BM = brain metastases; NR: not reported.

## Data Availability

Not applicable.

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
