# Peer review of "The Role of Molecular Imaging in Patients with Brain Metastases: A Literature Review"

_cancers, 2023, doi:10.3390/cancers15072184_

Round 1

Reviewer 1 Report

The authors provide an overview. on. current molecular imaging in patients with brain metastases and present future perspectives with new PET radiotracers,  their role, and potential approaches. 

I find the information well documented and summarized with updated literature and data collection.  

I miss in this review to extend a little bit more about future clinical trials to assessing new therapies with BM. 

English needs to be reviewed. 

Author Response

The authors provide an overview on current molecular imaging in patients with brain metastases and present future perspectives with new PET radiotracers, their role, and potential approaches.

I find the information well documented and summarized with updated literature and data collection.

R: We thank the reviewer for the appreciable comment.

I miss in this review to extend a little bit more about future clinical trials to assessing new therapies with BM.

R: We agree with the reviewer, and we have now included a last paragraph (6.4) focused on novel targeted therapies and relative PET imaging.

English needs to be reviewed.

R: We have read the manuscript and provided an English revision.

Reviewer 2 Report

This review summerized the role of molecular imaging in the evaluation of patients with brain metastases (BM), In the summary, the authors state that the  review summarizes the current use of positron emission tomography (PET) radiotracers in patients with BM, ranging from the present to the future perspectives with new PET radiotracers, including the role of radiomics and potential theranostics approaches. This is a bit misleading with only 1 small paragraph (6.3) on this without any data. So the summary needs to be rewritten.

method: no search strategy is provided. Also, it seems that important studies on this topic are missing, for example: DOI: 10.2967/jnumed.123.265420

doi:10.1093/annonc/mdv577, DOI: 10.1158/1078-0432.CCR-22-2720, 

doi: 10.1038/s41591-022-02084-8. etc

Also, the paper would gain interest if,  clinical relevant,  conclusions would be drawn in which settings molecular imaging is of value and in which settings we need further data to draw these conclusions. 

Author Response

This review summerized the role of molecular imaging in the evaluation of patients with brain metastases (BM), In the summary, the authors state that the review summarizes the current use of positron emission tomography (PET) radiotracers in patients with BM, ranging from the present to the future perspectives with new PET radiotracers, including the role of radiomics and potential theranostics approaches. This is a bit misleading with only 1 small paragraph (6.3) on this without any data. So the summary needs to be rewritten.

R: We agree with the reviewer that 6.3 is a small paragraph without data, although we have only underscored this aspect as “potential” for treatment of BM. However, we have re-phrased the last part of the introduction as follows: In this narrative review, we provide an overview on the current role of molecular imaging in patients with BM and briefly present future perspectives with new PET radiotracers, the role of radiomics and the newer targeted therapies. Finally, we open a suggestion for a potential application of theranostic approaches in BM, although clinical data are still missing.

method: no search strategy is provided. Also, it seems that important studies on this topic are missing, for example:

DOI: 10.2967/jnumed.123.265420

doi:10.1093/annonc/mdv577,

DOI: 10.1158/1078-0432.CCR-22-2720, 

doi: 10.1038/s41591-022-02084-8. Etc

R: We have included the research string as follows: The search string was (positron emission tomography[MeSH Terms]) AND (brain metastas*[Title/Abstract])”. However, this is a narrative and not a systemic review. In addition, we have reported the above mentioned papers in the last section (i.e. “future perspectives”)

Also, the paper would gain interest if, clinical relevant, conclusions would be drawn in which settings molecular imaging is of value and in which settings we need further data to draw these conclusions.

R: Thanks for the comment. We have reformulated our conclusions as follows: “To summarize, our review illustrates the role of molecular imaging in patients with BM. Currently, MRI represents the gold standard for diagnosis of BM, but PET is a valid complementary technique providing important biological findings that cannot be obtained from anatomical MRI alone. Amino acid PET radiotracers are the major representative of nuclear medicine in neuro-oncology, in particular for the diagnosis and assessment of post-treatment changes where results are encouraging. Furthermore, a wide variety of other PET agents has been developed and proposed for investigating different biological processes or for assessing response to newer therapies in patients with BM. including also neuropathological validation of imaging findings. Finally, the introduction of hybrid systems, e.g. PET/MRI, may improve the diagnostic process as they allow to acquire numerous multimodal imaging parameters in a shorter time, although at higher costs. However, large, prospective clinical trials are necessary to confirm such preliminary positive impressions in these cohort of patients”